# Intact protein folding in the glutathione-depleted endoplasmic reticulum implicates alternative protein thiol reductants

Satoshi Tsunoda[1,2,4], Edward Avezov[1,2,4], Alisa Zyryanova[1,2,4], Tasuku Konno[1,2,4], Leonardo Mendes-Silva[3], Eduardo Pinho Melo[3], Heather P Harding[1,2,4]*, David Ron[1,2,4]*

[1]Cambridge Institute for Medical Research, University of Cambridge, Cambridge, United Kingdom; [2]Wellcome Trust MRC Institute of Metabolic Science, Cambridge, United Kingdom; [3]Centre for Molecular and Structural Biomedicine, Universidade do Algarve, Faro, Portugal; [4]NIHR Cambridge Biomedical Research Centre, Cambridge, United Kingdom

**Abstract** Protein folding homeostasis in the endoplasmic reticulum (ER) requires efficient protein thiol oxidation, but also relies on a parallel reductive process to edit disulfides during the maturation or degradation of secreted proteins. To critically examine the widely held assumption that reduced ER glutathione fuels disulfide reduction, we expressed a modified form of a cytosolic glutathione-degrading enzyme, ChaC1, in the ER lumen. ChaC1$^{CtoS}$ purged the ER of glutathione eliciting the expected kinetic defect in oxidation of an ER-localized glutathione-coupled Grx1-roGFP2 optical probe, but had no effect on the disulfide editing-dependent maturation of the LDL receptor or the reduction-dependent degradation of misfolded alpha-1 antitrypsin. Furthermore, glutathione depletion had no measurable effect on induction of the unfolded protein response (UPR); a sensitive measure of ER protein folding homeostasis. These findings challenge the importance of reduced ER glutathione and suggest the existence of alternative electron donor(s) that maintain the reductive capacity of the ER.

*For correspondence: hph23@ cam.ac.uk (HPH); dr360@ medschl.cam.ac.uk (DR)

## Introduction

Disulfide bonds have a critical role in stabilizing correctly folded secreted and membrane proteins (*Fass, 2012*). Dedicated enzymatic machinery, consisting of disulfide exchange catalysts of the protein disulfide isomerase (PDI) class, rapidly introduces disulfide bonds into nascent polypeptide chains. The recycling of PDIs back to their oxidized form is mediated by upstream oxidases, exemplified by ERO1 and back-up enzymes, such as PRDX4 and VKOR that exploit $O_2$, $H_2O_2$, and vitamin K epoxide, respectively, as downstream electron acceptors. Defects in components of this electron transfer chain markedly affect protein folding homeostasis in the ER (*Sevier and Kaiser, 2008*; *Rutkevich and Williams, 2012*; *Zito, 2013*).

Many disulfides introduced by the oxidative machinery are not native and must be broken down in an editing process that is catalyzed by reduced forms of PDIs (*Hatahet and Ruddock, 2009*). This editing entails attack by the reduced N-terminal cysteine of PDI's active site on the misplaced disulfide following resolution of the mixed disulfide. The latter proceeds either by way of a zero-sum re-shuffling process entailing an attacking second reduced cysteine in the client polypeptide, which releases reduced PDI and establishes a new disulfide, or by a net reductive process involving an attack on the

**eLife digest** Proteins are basically strings of amino acids that have folded into a specific three-dimensional shape, and this shape is often important for the protein's function. Some proteins have bonds between pairs of cysteines—an amino acid that contains sulfur—in different parts of the protein to maintain its correct shape.

In eukaryotes, such as plants and animals, these so-called 'disulfide bonds' are formed inside a structure within each cell called the endoplasmic reticulum, which is where many proteins are folded. Occasionally, disulfide bonds form in the wrong place in a protein, so they need to be broken and re-positioned—a process sometimes called editing—for the protein to fold correctly. It was widely assumed that a chemical called 'reduced glutathione' fuels the breaking of disulfide bonds in the endoplasmic reticulum, but to date few researchers have tried to test this assumption.

Tsunoda et al. have now taken an enzyme that degrades glutathione elsewhere in the cell and modified it in a way that allows it to work inside the endoplasmic reticulum. When this modified enzyme was produced in human cells grown in the laboratory, it purged the endoplasmic reticulum of glutathione. However, the lack of glutathione had no effect on the folding of a large protein with 30 disulfide bonds, many of which need to be edited at one time or another for the protein to fold correctly. The destruction of a poorly folded protein, via a process that also needs this protein's disulfide bonds to be broken down, was also not affected by a lack of reduced glutathione in the endoplasmic reticulum.

Furthermore, decreasing these levels of glutathione did not affect the unfolded protein response: a stress response in cells that are experiencing a build-up of unfolded or poorly folded proteins within the endoplasmic reticulum.

As such, the findings of Tsunoda et al. challenge the importance of reduced glutathione in the endoplasmic reticulum and suggest that other chemical processes might be involved in editing disulfide bonds. Further work is now needed to investigate the other known processes that might complete this task instead to see which, if any, are involved.

mixed disulfide by the C-terminal active site cysteine, in which PDI 'escapes' in its oxidized form, having reduced its client protein (**Walker et al., 1996**). Both mechanisms require a pool of reduced PDI, the maintenance of which is not trivial, given the oxidative environment in the ER. This challenge is especially great in case of the second mechanism, which is a set-up for iterative cycles of net oxidation-reduction-oxidation (**Schwaller et al., 2003**).

The crucial role of reduced PDI to oxidative protein folding is supported by in vitro experiments in which PDI's ability to accelerate folding is not monotonically increased by its oxidized fraction, but rather is optimal in a redox buffer that also contains a high concentration of reduced glutathione (**Lyles and Gilbert, 1991**). Yeast genetics reveal that disulfide shuffling (which requires reduced PDI) is *PDI1*'s essential function, as an active-site mutant that has lost the ability to produce disulfides (but has selectively retained activity as a disulfide reductase) is nonetheless able to rescue the lethal phenotype of *pdi1* nullizygosity (**Laboissiere et al., 1995**).

The reductive facet of oxidative protein folding in the ER is especially important to the maturation of large proteins such as the low density lipoprotein receptor (LDL-R), in which it has been estimated that most disulfides that form early during biogenesis are non-native and must be rearranged before the protein clears ER quality control and traffics to the Golgi (**Jansens et al., 2002**). This editing process appears to involve a specific PDI family member, ERdj5 (**Oka et al., 2013**). ERdj5 may have specialized in disulfide reduction, as its redox function also accelerates the clearance of misfolded ER proteins such as the null Hong Kong mutant α1-antitrypsin (NHK-A1AT) (**Ushioda et al., 2008**; **Hagiwara et al., 2011**), but the identity of ERdj5's reductase remains unknown.

Reductive editing of disulfides is also observed in the *E. coli* periplasm. Where the transfer of electrons from reduced thioredoxin in the cytosol maintains a reduced pool of the periplasmic isomerases, DsbC and DsbG. Electrons are conveyed across the inner-membrane space by a specialized transmembrane protein DsbD. This protein relay-based mechanism enables DsbC/DsbG-dependent disulfide shuffling despite the absence of a soluble small molecule redox buffer in the periplasmic space of gram negative bacteria (reviewed in **Cho and Collet, 2013**). By contrast, the mammalian ER contains

up to 15 mM glutathione (*Montero et al., 2013*) whose reduced form is widely believed to fuel the reductive aspects of secreted protein metabolism in eukaryotes, by serving as a terminal electron donor to reduce PDI family members (reviewed in *Kojer and Riemer, 2014*).

To critically examine the role of ER glutathione in the reductive re-shuffling of non-native disulfides and in the reductive steps believed to be associated with degradation of misfolded ER proteins, we devised a method to selectively deplete the ER of glutathione and examined the consequences on the organelles' capacity to handle well-characterized sentinel proteins.

## Results

### A glutathione-degrading enzyme suited to the oxidizing conditions of the endoplasmic reticulum

Kumar et al. recently reported that the mammalian pro-apoptotic gene *ChaC1* encodes a glutathione-specific γ-glutamyl cyclotransferase that efficiently degrades glutathione (*Kumar et al., 2012*). We confirmed their observations by measuring the ability of purified murine ChaC1 (expressed in *E. coli*) to degrade glutathione in vitro: At submicromolar enzyme concentrations, recombinant ChaC1 was able to degrade a 10 mM solution of reduced glutathione within 1 hr (*Figure 1A*). The enzymatic activity was selective towards reduced glutathione (GSH) (*Figure 1B*). The inability of oxidized glutathione (GSSG) to serve as a substrate for degradation correlated with its inability to bind a Chac1-based optical probe whose fluorescent resonance energy transfer (FRET) signal reflects substrate binding (*Figure 1C–D* and *Figure 1—figure supplement 1*). An E116Q mutation abolished all enzymatic activity (*Figure 1E*), as observed previously (*Kumar et al., 2012*).

To exploit ChaC1 as a tool to purge the ER of glutathione, we targeted expression of this cytosolic enzyme to the ER, by fusing the coding sequence to an N-terminal cleavable signal peptide and a C-terminal KDEL ER retention signal. An N-terminal FLAG-M1 peptide tag was included, to facilitate detection of the enzyme. Cells transfected with a plasmid encoding ER-FLAG-ChaC1 expressed a protein of the expected mobility on reducing SDS-PAGE that reacted with the anti-FLAG antibody (*Figure 2A*) and resulted in a granular staining pattern that overlapped with that of the ER marker calreticulin (*Figure 2B*). However, unlike cytosolic ChaC1, which migrates at a position expected of the reduced monomer on non-reducing SDS-PAGE (*Figure 2—figure supplement 1*), ER-localized ChaC1 migrated as a heterogenous collection of species, consistent with inappropriate disulfide bond formation (compare the reducing and non-reducing SDS-PAGE in *Figure 2A* and *Figure 2—figure supplement 1*).

A model of mouse ChaC1, based on the crystal structure of the homologous γ-glutamyl cyclotransferase (PDB 2RBH) (*Kumar et al., 2012*), suggested that none of its four cysteines is likely to play an important role in substrate recognition or catalysis. As expected, conversion of all four cysteines to serines resulted in a protein that no longer formed disulfides when introduced into the ER (*Figure 2A*, compare lanes 6 and 7 and *Figure 2—figure supplement 1*). Importantly, the cysteine-free enzyme (ChaC1^CtoS) purified from *E. coli* retained its enzymatic activity (*Figure 2C*) and specificity for reduced glutathione (*Figure 1D* and *Figure 1—figure supplement 1C*). There are no predicted N-linked glycosylation sites in ChaC1^CtoS to further corrupt protein structure when targeted to the ER, therefore, it seemed possible that ER-localized ChaC1^CtoS might retain its enzymatic activity and breakdown glutathione in the ER.

### Purging the ER of glutathione

Measuring the impact of ER-ChaC1^CtoS on glutathione levels required an assay that would be selectively sensitive to the ER pool of glutathione. Glutaredoxin (Grx1) has been shown to dramatically accelerate the interaction of a linked redox-sensitive green fluorescent protein (roGFP) with glutathione, both in vivo and in vitro (*Gutscher et al., 2008*; *Birk et al., 2013*) (cartooned *Figure 3A*). We confirmed the reported ability of a linked Grx1 to accelerate the equilibration of roGFP with a glutathione buffer: alone, reduced roGFP2 was only slowly oxidized by glutathione (*Figure 3B*), but the linked Grx1 markedly accelerated the oxidation of Grx1-roGFP2 (*Figure 3*, compare the red traces in panels B and C). The rate of probe oxidation by glutathione was concentration-dependent, with half-saturation ($Kmax_{0.5}$) attained at ~18 μM GSSG (*Figure 3D*). Importantly, the presence of oxidized PDI had a minor role in further accelerating the oxidation of Grx1-roGFP2, but dominated the oxidation kinetics of roGFP2 alone (*Figure 3B,C*).

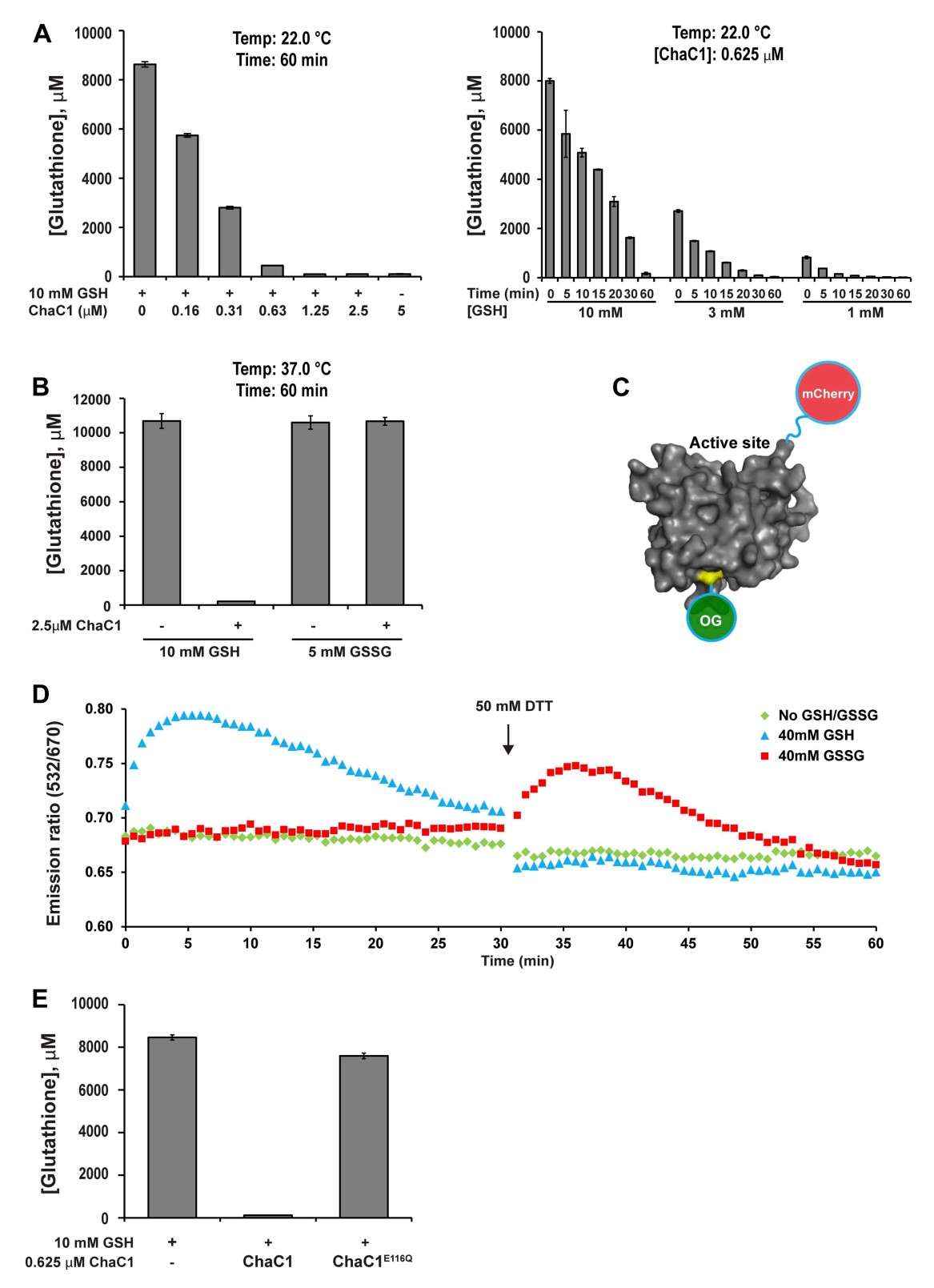

**Figure 1**. ChaC1 efficiently and selectively degrades reduced glutathione. (**A**) A bar-graph representation of residual glutathione levels following incubation of 10 mM glutathione with the indicated concentrations of bacterially expressed mouse ChaC1. Varying concentrations of enzymes were assayed at a single time point (left panel) and varying initial concentrations of glutathione were assayed at different time points (right panel). (**B**) Comparison

*Figure 1. Continued on next page*

*Figure 1. Continued*

of the ability of ChaC1 to eliminate reduced (GSH) and oxidized glutathione (GSSG). Note that ChaC1 effectively eliminated reduced glutathione, but had no effect on oxidized glutathione. (**C**) Cartoon of the fluorescent resonance energy transfer (FRET) probe, OG-ChaC1-Cherry, used to detect substrate binding to ChaC1. Shown is a model of murine ChaC1 (UniProt Q8R3J5) residues 31–204, created by Phyre[2] (*Kelley and Sternberg, 2009*) based on the crystal structure of γ-glutamyl cyclotransferase (PDB 2RBH). The side chain of Cys 92, which has been modified with the Oregon Green (OG) donor, is highlighted, as is the C-terminus of the protein, site of the fused mCherry fluorescent acceptor. (**D**) Time-resolved FRET signal (expressed as the ratio of the emission signal at 532 nm and 670 nm upon excitation at 480 nm) of the OG-ChaC1-Cherry probe [2.5 µM] following exposure to 10 mM reduced (GSH) or oxidized glutathione(GSSG). Where indicated, the sample was injected with dithiotreitol (DTT) to reduce the GSSG and convert it to a substrate for ChaC1. The biphasic change in FRET signal upon exposure to GSH is consistent with binding followed by breakdown of GSH by the probe, which retains its enzymatic activity. (**E**) Comparison of glutathione elimination by purified bacterially expressed wild-type and E116Q mutant ChaC1 in vitro.

The following figure supplements are available for figure 1:

**Figure supplement 1**. Analysis of the substrate binding properties of ChaC1.

Oxidized glutathione levels recover rapidly following a reductive pulse in cultured mammalian cells (*Appenzeller-Herzog et al., 2010*). Therefore, the aforementioned in vitro observations indicating tight coupling of Grx1-roGFP2 to glutathione (and relative indifference to PDI) suggested that the rate of re-oxidation of an ER-localized Grx1-roGFP2 probe following a reductive pulse might be affected by the presence of glutathione in the ER lumen. To examine this further, we first compared the rate of re-oxidation of ER-localized roGFP2 with that of ER-localized Grx1-roGFP2 following a reductive pulse of dithiothreitol (DTT) and its washout. At steady-state both probes were highly oxidized and both were similarly reduced by the DTT pulse. However the recovery of ER-Grx1-roGFP2 was accelerated compared with ER-roGFP2 alone, with a half-time to recovery of 75.3 ± 7.7 s in case of the former and 167.3 ± 16.8 s in case of the latter (*Figure 3E,F*). This observation is consistent with a direct contribution of lumenal glutathione to the kinetics of ER-Grx1-roGPF2 re-oxidation in vivo.

Next, we compared the effect of ER-localized enzymatically-active ER-ChaC1[CtoS] and its catalytically dead counterpart, ER-ChaC1[CtoS–E116Q] on the rate of oxidation of ER-roGFP2 and ER-Grx1-roGFP2 following a reductive pulse and washout. To focus the measurements on cells co-expressing the roGFP probe and the ChaC1 enzyme, the latter was tagged at its C-terminus with mCherry. Oxidation of ER-roGFP2 was unaffected by the presence of active ChaC1, but the kinetic advantage of ER-Grx1-roGFP2 over the ER-roGFP2 alone was abolished by expression of an active, glutathione-degrading enzyme in the ER lumen (*Figure 3G,H*).

We were unable to reproducibly measure glutathione concentrations in microsomal fractions of cultured cells; however, the effect of ChaC1 over-expression on total cellular glutathione was quantifiable. At similar levels of over-expression, cytosolic ChaC1 led to a marked depletion of total cellular glutathione levels, whereas ER-localized ChaC1[CtoS] had a more modest effect. However, ER-localized ChaC1[CtoS] markedly enhanced glutathione depletion by low concentrations of buthionine-sulfoxide (BSO, an inhibitor of the rate limiting step of glutathione biosynthesis, *Figure 3I*). To avoid the corrupting effect of an untransfected pool of cells, these ensemble measurements were conducted in stable clones homogenously and conditionally-expressing ChaC1 from a doxycycline-inducible transgene. They point to relatively slow equilibration of cytosolic and ER pools of glutathione in mammalian cells. Given that Grx1-roGFP2 reacts with glutathione with a $Kmax_{0.5}$ in the $10^{-5}$ M range (*Figure 3D*) the effacement of its kinetic advantage over roGFP2 in the DTT washout experiment, indicated a profound and selective depletion of lumenal glutathione by the ER-targeted expression of active ChaC1, with modest effects on other cellular pools of glutathione.

## Lumenal glutathione is dispensable to the reductive facets of ER protein folding

Maturation of the LDL-R in the ER entails significant rearrangement of its 30 disulfide bonds (*Jansens et al., 2002*) and involves the reduced form of the PDI family member ERdj5 (*Oka et al., 2013*). Maturation of the LDL-R can be tracked by the conversion of the relatively high mobility, glycosylated, ER form, to the lower-mobility post-ER form (reflective of Golgi sugar modifications) in pulse-chase labeling followed by immunoprecipitation. We thus compared the effects of ER expression of active ChaC1[CtoS] and the catalytically inactive ChaC1[CtoS;E116Q] mutant on the rate of maturation of co-expressed LDL-R tagged on its C-terminal, cytosolic facing domain, with a triple FLAG-tag.

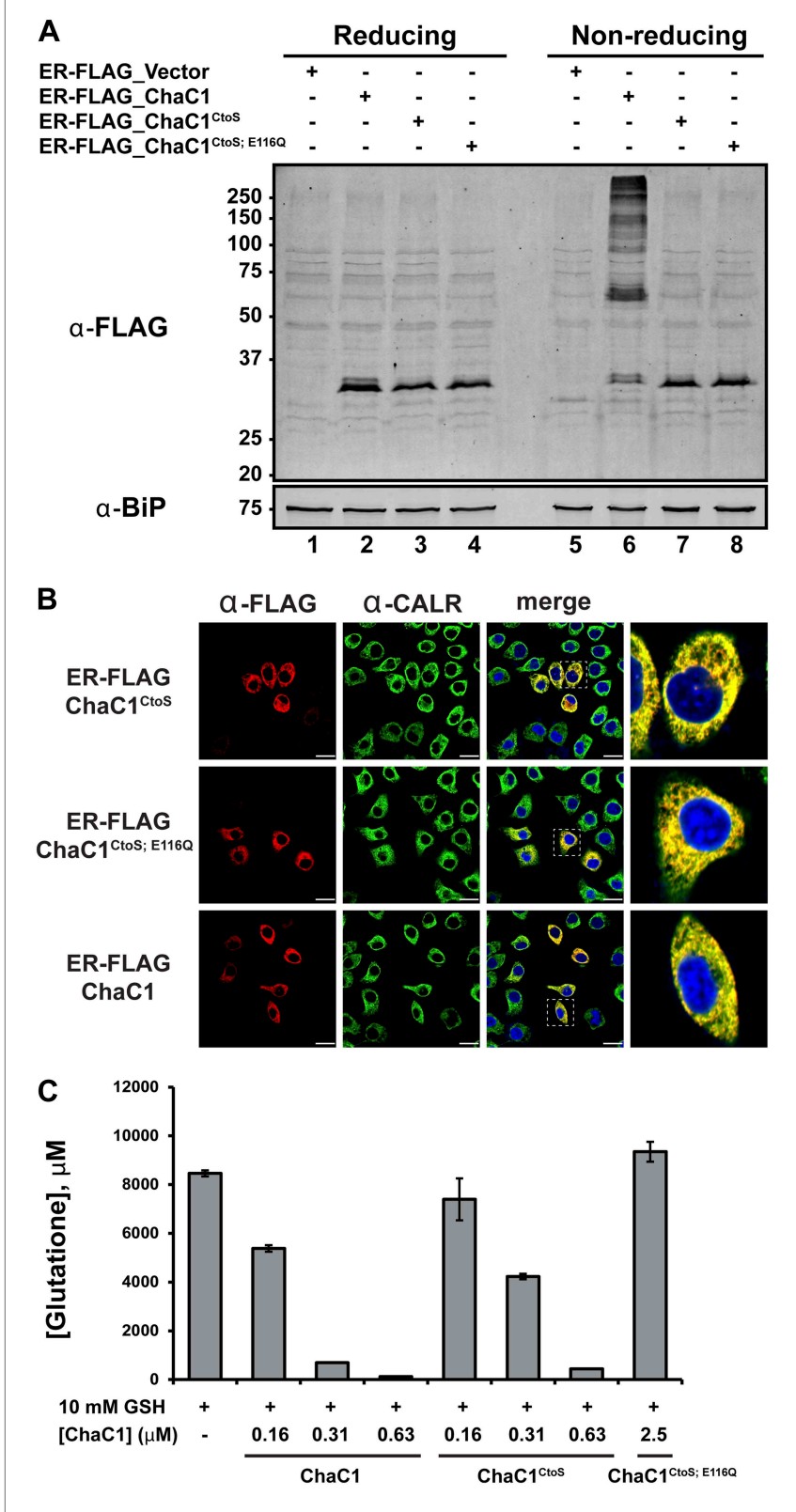

**Figure 2**. Cysteine-free ChaC1 is an active enzyme that can be targeted to the endoplasmic-reticulum.
(**A**) Immunoblot of FLAG M1-tagged ER-localized wild type (WT), cysteine-free (CtoS), and compound CtoS; E116Q
enzymatically dead ChaC1 in N-ethyl maleimide-blocked lysates of transfected HeLa cells. Lanes 1–4 are from a
*Figure 2. Continued on next page*

*Figure 2. Continued*

reducing and lanes 5–8 are from a non-reducing SDS-PAGE. Note the presence of high-molecular weight disulfide linked FLAG-tagged ChaC1 in cells transfected with the WT ER-targeted protein that is absent from those transfected with the cysteine-free, CtoS mutants. (**B**) Fluorescent photomicrographs of HeLa cells transfected with the indicated expression plasmids and immunostained for the FLAG tag (marking ChaC1) and calreticulin (CALR) as an ER marker. The merge panels show an overlap of the FLAG, CALR, and Hoechst 33,258 signal (to reveal the nuclei) at 630X with a close-up view in the right-most panel. (**C**) A bar-graph representation of residual glutathione levels following 60-min incubation of 10 mM glutathione with the indicated concentrations of bacterially expressed wild-type ChaC1, its cysteine free derivative, ChaC1$^{CtoS}$, and its inactive mutant ChaC1$^{CtoS;E116Q}$.

The following figure supplements are available for figure 2:

**Figure supplement 1**. Replacement of cysteines with serines circumvents aberrant disulfide bond formation in ER-localized ChaC1.

Conversion of the LDL-R from its ER to post-ER form was unaffected by the presence of either active or inactive form of ER-ChaC1 and proceeded with a half-life of ~1 hr in presence or absence of an active glutathione-depleting enzyme in the ER (*Figure 4A*). The non-reducing gel revealed identical accelerated mobility of LDL-R at the earliest time point, regardless of expression of ChaC1, indicating that glutathione depletion also had no drastic effect on the earlier oxidative phase of LDL-R maturation (*Figure 4A,B*).

LDL-R maturation was also unaffected by further global depletion of cellular glutathione, effected by the combined expression of ER-ChaC1$^{CtoS}$ and exposure to buthionine sulfoxide (BSO), which depleted glutathione globally (*Figure 4C–E*). These experiments, conducted in HeLa cells, where reductive editing of LDL-R disulfides was first discovered (*Jansens et al., 2002*), reinforces the dispensability of glutathione to that process.

To further explore the potential role of ER glutathione in the early oxidative steps of protein folding, we used highly sensitized mouse cells genetically deficient in the upstream thiol oxidases ERO1 and PRDX4 (*Zito et al., 2010b*, *2012*) Expression of active ChaC1 in their ER had no effect on the kinetics of disulfide re-formation on ER-localized roGFP2 following a reductive DTT pulse (*Figure 4F–G*), further attesting to the dispensability of glutathione to thiol oxidation in the mammalian ER.

The degradation of misfolded NHK-A1AT requires the action of the reduced form of a specialized PDI, ERdj5 (*Ushioda et al., 2008*; *Hagiwara et al., 2011*). To determine if an ER pool of glutathione contributes to this process, we compared the half-life of C-terminally FLAG-tagged NHK-A1AT in cells expressing active ER-ChaC1$^{CtoS}$ and the catalytically inactive ER-ChaC1$^{CtoS;E116Q}$. The stability of NHK-A1AT was unaffected by ChaC1 both in HeLa cells (*Figure 5A,D*) and in 293T cells (*Figure 5B,E*). Furthermore, depletion of total cellular pools of glutathione by coincidental exposure to BSO had no effect on NHK-A1AT half-life (*Figure 5C,F*), complementing the evidence for the dispensability of glutathione for ER-associated degradation of this redox-dependent substrate.

Thiol redox reactions contribute to protein folding homeostasis in the endoplasmic reticulum. This is reflected in enhanced signaling in the endoplasmic reticulum unfolded protein response (UPR) in cells impaired in ER thiol redox (*Frand and Kaiser, 1998*; *Pollard et al., 1998*). Therefore, to gain a more global view on the impact of ER glutathione depletion on ER protein folding homeostasis, we compared the effect on UPR activity of ER-targeted active and inactive ChaC1. These experiments made use of mCherry-tagged ChaC1$^{CtoS}$, which retains its enzymatic activity, purges the ER of glutathione (*Figures 1D and 3G*) and marks the ChaC1-expressing cells.

Dual-channel FACS analysis revealed that neither active nor inactive ER-ChaC1$^{CtoS}$-mCherry-KDEL measurably affected the basal activity of a stably integrated mammalian UPR reporter, CHOP::GFP (*Novoa et al., 2001*) (reflected in the absence of a shift to the right in the mCherry positive population of cells, *Figure 6A*, left column). Furthermore, the activity of the CHOP::GFP reporter, which was increased by tunicamycin, a toxin that perturbs protein folding homeostasis in the ER, was unaffected by ChaC1 (*Figure 6A*). The indifference of the UPR to ER-ChaC1$^{CtoS}$ is observed over a broad range of unfolded protein stress and over a broad range of ER-ChaC1$^{CtoS}$-mCherry-KDEL expression, and suggests that the lesson learned from the sentinel proteins, LDL-R and NHK-A1AT likely extend to the bulk of proteins that fold oxidatively in the ER under normal cell culture conditions.

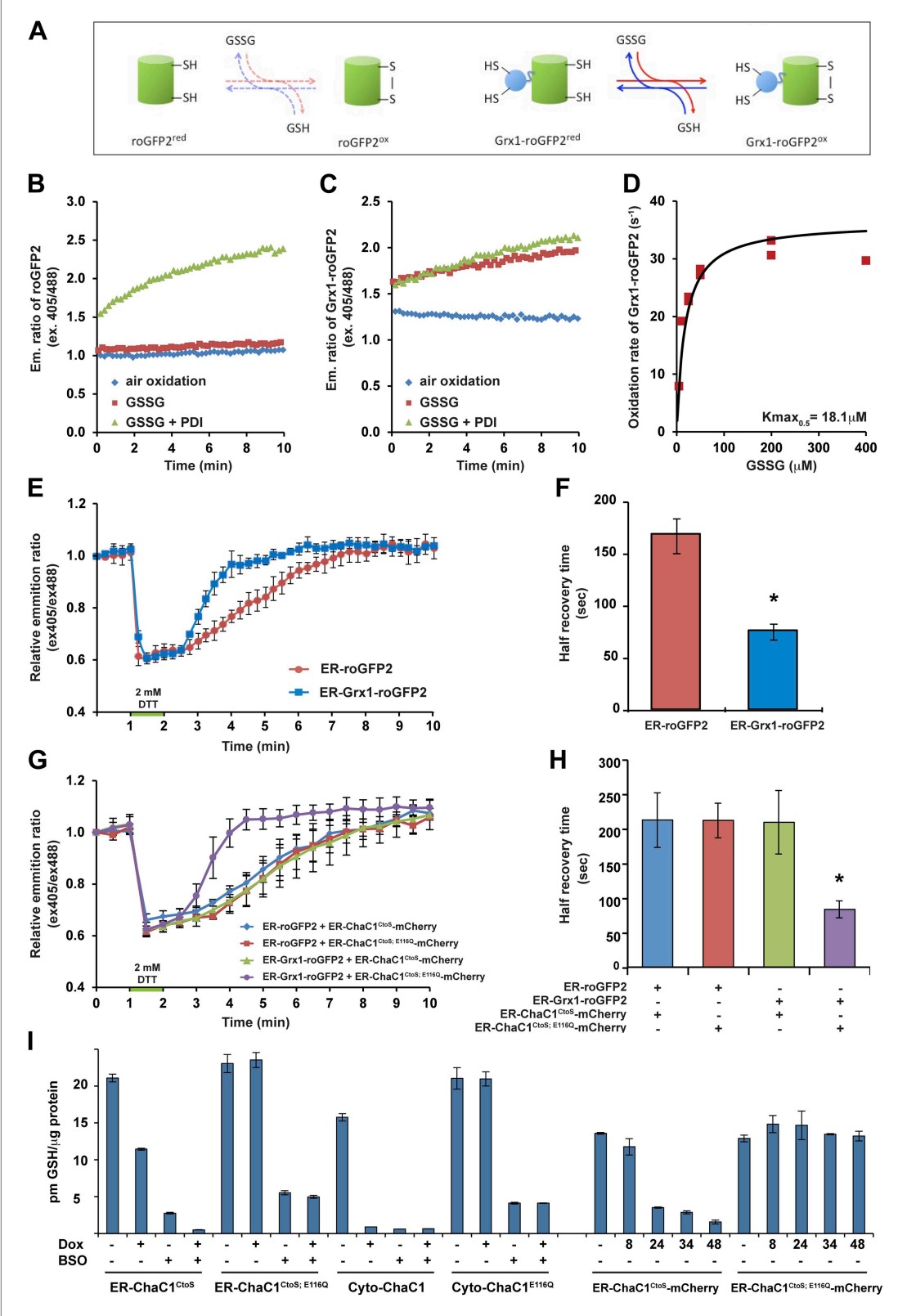

**Figure 3**. ER-targeted ChaC1 purges the organelle of its glutathione content. (**A**) Cartoon contrasting the slow coupling of roGFP2 with the glutathione redox buffer (dashed lines) and the rapid coupling of Grx1-roGFP2 with the glutathione redox buffer (after *Gutscher et al., 2008*). (**B**) Trace of time-dependent changes in the ratio of reduced to oxidized roGFP2 detected optically as the ratio between the emission signal (at 535 nm) upon excitation
*Figure 3. Continued on next page*

*Figure 3. Continued*

at 405 nm vs 488 nm (ex 405/488) following introduction of the fully reduced probe into the indicated solutions of oxidized glutathione (GSSG) or PDI and GSSG. (**C**) Similar trace of reduced Grx1-roGFP2. Note the indifference of roGFP2 and the marked responsiveness of Grx1-roGFP to oxidized glutathione. (**D**) Graph of the initial velocity of Grx1-roGFP2 oxidation as a function of GSSG concentration, fitted to Linweaver–Burk plot. Half-maximal velocity is observed at 18 µM GSSG. (**E**) Trace of time-dependent changes in the ratio of oxidized and reduced roGFP2 and Grx1-roGFP2 probes expressed in the ER of HeLa cells following a brief (1 min) reductive pulse with dithiothreitol (DTT, 2 mM) followed by a washout. (**F**) Bar diagram of the half-time to recovery of oxidized roGFP2 and Grx1-roGFP2 following the reductive DTT pulse. Shown are means ±SD (N = 4, *p<0.01). (**G**) Trace of time-dependent changes in the ratio of oxidized to reduced probes expressed in the ER of HeLa cells alongside active or inactive ChaC1 (tagged at its C-terminus with mCherry to allow visualization of cells co-expressing the redox probes and the glutathione-depleting enzyme) following a brief reductive pulse with dithiothreitol and a washout. Note that the expression of active ChaC1 in the ER eliminates the kinetic advantage of Grx1-roGFP2 over roGFP2 in re-oxidation during the recovery from a DTT reductive pulse. (**H**) Bar diagram of the half-time to recovery of oxidized roGFP2 and Grx1-roGFP2 following the reductive DTT pulse in cells co-expressing active or inactive CHaC1. Shown are means ±SEM (N = 20, *p<0.01). (**I**) Bar diagram of cellular glutathione levels following 36 hr of doxycycline (DOX) induction of cytosolic and ER localized active and inactive ChaC1 in the absence and presence of concomitant exposure to buthionine-sulfoxide (BSO, 50 µM). Also shown is a time course of total cellular glutathione following induction of active and inactive mCherry-KDEL-tagged ChaC1.

The role of ER glutathione was further explored under conditions in which ER redox balance was perturbed by co-expression of a deregulated allele of the ER oxidase ERO1 (*Sevier et al., 2007*), which has been shown to hyper-oxidize the mammalian ER (*Baker et al., 2008*) and modestly activate the UPR (*Hansen et al., 2012*). Introduction of the C104A; C133A human ERO1L (ERO1*), expressed from a plasmid tagged with the CD2 surface marker indeed modestly activated the UPR, whether measured in cells stably expressing the CHOP::GFP transcriptional reporter (*Figure 6B*, reflected in the higher GFP levels of cells co-expressing the CD2 marker which tags the ERO1*-expressing cells, panels 2, 4, and 7) or an XBP1::Venus splicing reporter (*Iwawaki et al., 2004*) (*Figure 6—figure supplement 1*). However, co-expression of active ChaC1[CtoS] had no evident synergistic effect with ERO1* on UPR activity beyond that observed with the enzymatically inactive E116Q mutant enzyme. This is evident by first noting that in this co-transfection experiment most ERO1* expressing cells (CD2 positive) are also co-expressing the Cherry-tagged ChaC1[CtoS] (*Figure 6B*, panels 3 and 6) and then noting that the relationship between ERO1* expression (for which CD2 is a surrogate) and the CHOP::GFP signal is indistinguishable in cells co-expressing wild-type and enzymatically inactive ChaC1[CtoS] (*Figure 6B* panels 4 and 7). Similarly, the co-expression of ERO1* does not impart sensitivity to active ChaC1, as reflected by the observation that CHOP::GFP levels are unaffected by ChaC1[CtoS] in these double positive cells (*Figure 6B* panels 5 and 8). These findings indicate that glutathione is dispensable for the function of the ER even under hyperoxidizing conditions.

## Discussion

Adapting a cytosolic enzyme that breaks down glutathione to function in the ER has afforded a means to selectively purge the organelle of glutathione and thereby assess the role of this otherwise abundant tri-peptide in ER protein redox. Nearly complete depletion of ER glutathione had no measurable effect on the rate of disulfide bond formation in the highly sensitized experimental setting of cells deficient in ER thiol oxidases. More surprising was the lack of any measurable effect of glutathione depletion on processes that require a pool of reduced PDI enzymes: the disulfide shuffling-dependent maturation of nascent LDL-R and the degradation of a folding incompetent mutant NHK-A1AT. Furthermore, depletion of ER glutathione was without effect on two sensitive, broad-spectrum UPR reporters, arguing that glutathione is indeed dispensable to a broad range of processes required for protein folding homeostasis in the ER of cultured mammalian cells.

The dispensability of glutathione to reductive processes in the ER has been hinted at previously: in yeast deletion of *GSH1* (encoding gamma glutamylcysteine synthetase that performs the rate-limiting step in glutathione biosynthesis) does not adversely affect the maturation of the disulfide-bonded lysosomal hydrolyze CPY (*Frand and Kaiser, 1998*). However, the extent of ER glutathione depletion attained by this genetic manipulation is unclear and, given the essential role of cellular glutathione to yeast growth, may have been limited by toxicity. Similar reservations apply to the use of enzyme inhibitors, such as BSO, which subject the cell to the consequences of glutathione depletion in other compartments where it has an essential role (*Kumar et al., 2011*), but whose effect on the ER pools of glutathione may be partial.

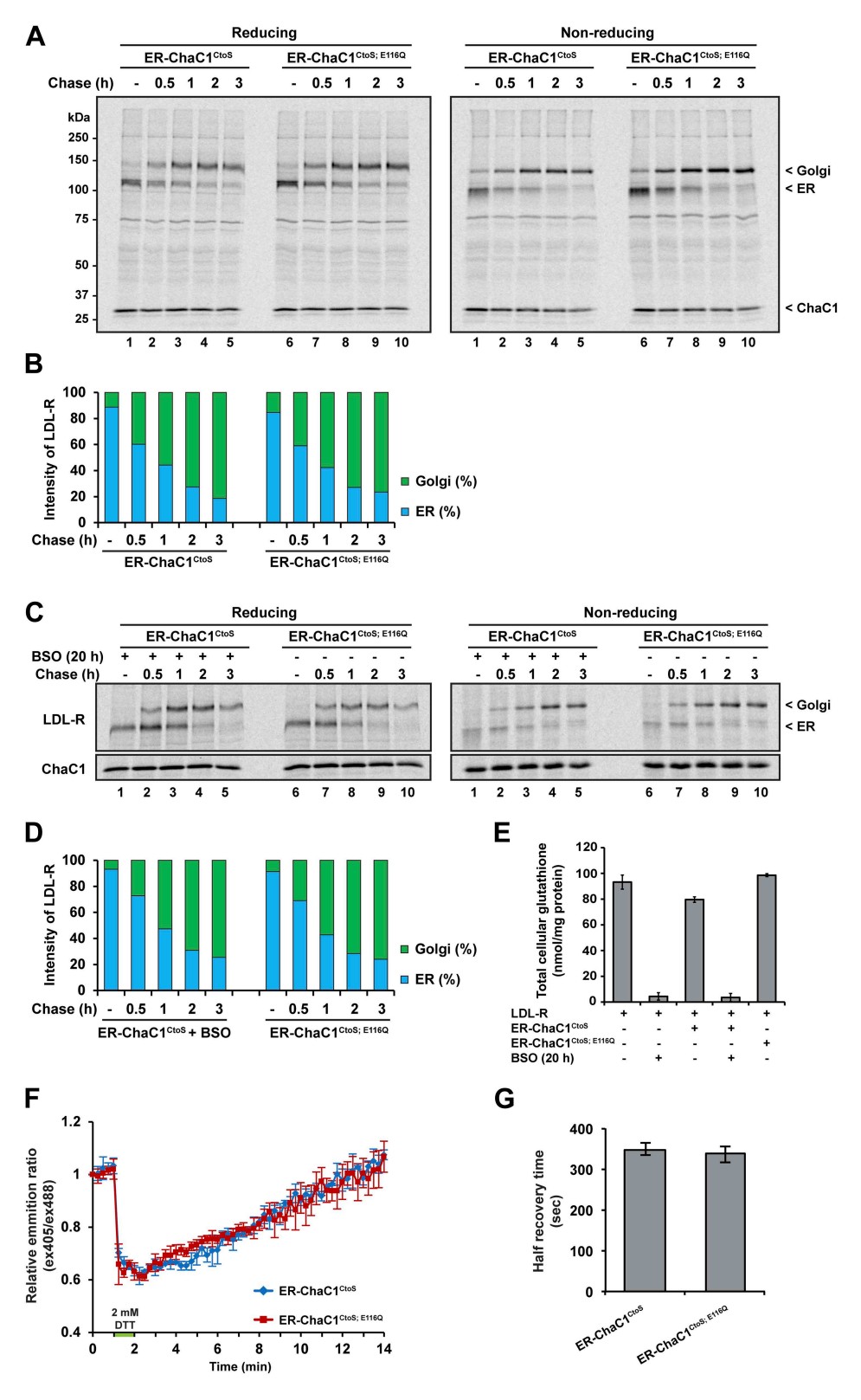

**Figure 4**. Maturation of the LDL-R receptor is unaffected by depletion of ER glutathione. (**A**) Autoradiograph of metabolically labeled LDL receptor (LDL-R) immunopurified from HeLa cells co-expressing the FLAG-tagged LDL-R and ER-localized enzymatically active (ER-ChaC1$^{CtoS}$) or inactive ChaC1 (ER-ChaC1$^{CtoS;E116Q}$) and resolved on a
*Figure 4. Continued on next page*

*Figure 4. Continued*

reducing SDS-PAGE. Cells were lysed at the end of a 30-min labeling pulse (lanes 1 and 6) or after an additional chase period (indicated). The mobility of the ER and Golgi forms of the LDL-R on reducing (left) and non-reducing (right) SDS-PAGE is indicated, as is the labeled ChaC1, which is also recovered in this anti-FLAG immunoprecipitation. (**B**) Graphic presentation of the conversion of the ER to Golgi form of the LDL-R from '**A**' above. Shown is a representative experiment reproduce three times with similar outcome. (**C**) Autoradiograph of an experiment similar in layout to that depicted in '**A**' above. Where indicated, the cells were exposed to the glutathione synthesis inhibitor buthionine-sulfoxide (BSO, 100 µM, 20 hr) before the pulse-chase labeling. (**D**) Graphic presentation of the conversion of the ER to Golgi form of the LDL-R from '**C**' above. Shown is a representative experiment reproduce three times with similar outcome. (**E**) Bar graph of total cellular glutathione in cells manipulated as in the experiment described in '**C**' (shown is the mean ±SD, n = 3, *p<0.05). (**F**) Trace of time-dependent changes in the ratio of oxidized and reduced roGFP2 expressed in the ER of mouse embryonic fibroblasts with genetic lesions compromising disulfide bond formation (Ero1a$^{m/m}$; Ero1b$^{m/m}$; Prdx4$^{m/y}$) following a brief reductive pulse with dithiothreitol followed by a washout. The cells co-expressed ER-localized enzymatically active (ER-ChaC1$^{CtoS}$) or inactive ChaC1 (ER-ChaC1$^{CtoS;E116Q}$) marked at its C-terminus with an mCherry fluorescent probe. (**G**) Bar diagram of the half-time to recovery of oxidized roGFP2 following the reductive DTT pulse in '**F**' above. Note that expression of active ChaC1 in the ER did not affect the rate of recovery of the sentinel disulfide in ER-localized roGFP2.

How complete was depletion of ER glutathione by ChaC1? In vitro the re-oxidation of Grx1-roGFP2 is half-maximal at 18 µM glutathione (*Figure 3D*). Thus, the abolition of the kinetic advantage of Grx1-roGFP2 over roGFP2 in cells expressing enzymatically active ChaC1 argues that, despite the enzyme's relative low affinity for its substrate (in the millimolar range, *Kumar et al., 2012*; and *Figure 1—figure supplement 1C,D*), glutathione levels were purged to micromolar levels by the high concentration of the enzyme. This conclusion is also supported by the effects of ER ChaC1 on total cellular glutathione levels: At similar levels of expression, cytosolic ChaC1 led to near complete depletion of total cellular glutathione, whereas ER ChaC1 had a more modest effect on total cellular pools. Given the 10-fold greater volume of the cytosol over the ER (*Stefan et al., 1987*), the aforementioned observations indicate substantially higher local concentration of ER vs cytosolic ChaC1. Thus, the preservation of total cellular glutathione levels in cells expressing high levels of ChaC1 in their ER confirms that transport of glutathione into the ER, from its site of synthesis in the cytosol, is slow (as suggested by *Kumar et al., 2011*), and therefore, that depletion of ER stores by ChaC1 is profound.

The redundancy of glutathione could be explained by other small molecule thiols fueling reduction in the ER. These may even include cysteinyl-glycine, one of the glutathione breakdown product of ChaC1 action. Alternatively, eukaryotes may have a protein-driven apparatus akin to bacterial DsbD for ferrying reducing equivalents from the cytosol to reduce PDI family members.

Despite its apparent dispensability, ER glutathione does equilibrate with protein thiols (*Appenzeller-Herzog et al., 2010*), indicating that it is a redox buffer in the organelle. However, it is possible that despite seemingly rapid equilibration of glutathione to protein thiol redox, the kinetics are insufficient to render glutathione essential and that other, even faster processes, maintain an adequate pool of reduced and oxidized PDIs in the mammalian ER, with glutathione following passively. It is noteworthy, in this vein, that ERO1-deficient cells, with compromised ER thiol oxidative power, have an elevated ratio of oxidized to reduced glutathione (*Appenzeller-Herzog et al., 2010*; *Rutkevich and Williams, 2012*). It has been speculated, reasonably, that this reflects the action of an alternative oxidative pathway that kicks-in when the major oxidases are compromised and exploits oxidized glutathione to couple to protein thiol oxidation. However, our finding that ER glutathione depletion does not further compromise protein thiol oxidation kinetics in oxidase-deficient cells suggests that here too glutathione follows passively the rearrangements in redox pathways and does not participate actively in their implementation.

Across distant phyla, ER redox is not indifferent to cellular glutathione levels. Depletion of glutathione, by *GSH1* deletion, rescues the oxidative defect in ERO1-deficient yeast (*Cuozzo and Kaiser, 1999*), and cytosolic glutathione influences disulfide-bonding of glutenin in the ER of wheat endosperm (*Lombardi et al., 2012*). These observations, which fall short of directly indicting glutathione in reducing ER disulfides, nonetheless argue for participation of glutathione in a shared economy of reducing equivalents across the ER membrane. Prolonged (48 hr) depletion of glutathione by high concentrations of BSO (1 mM) synergize with ER hyperoxidation to degrade cell viability (*Hansen et al., 2012*).

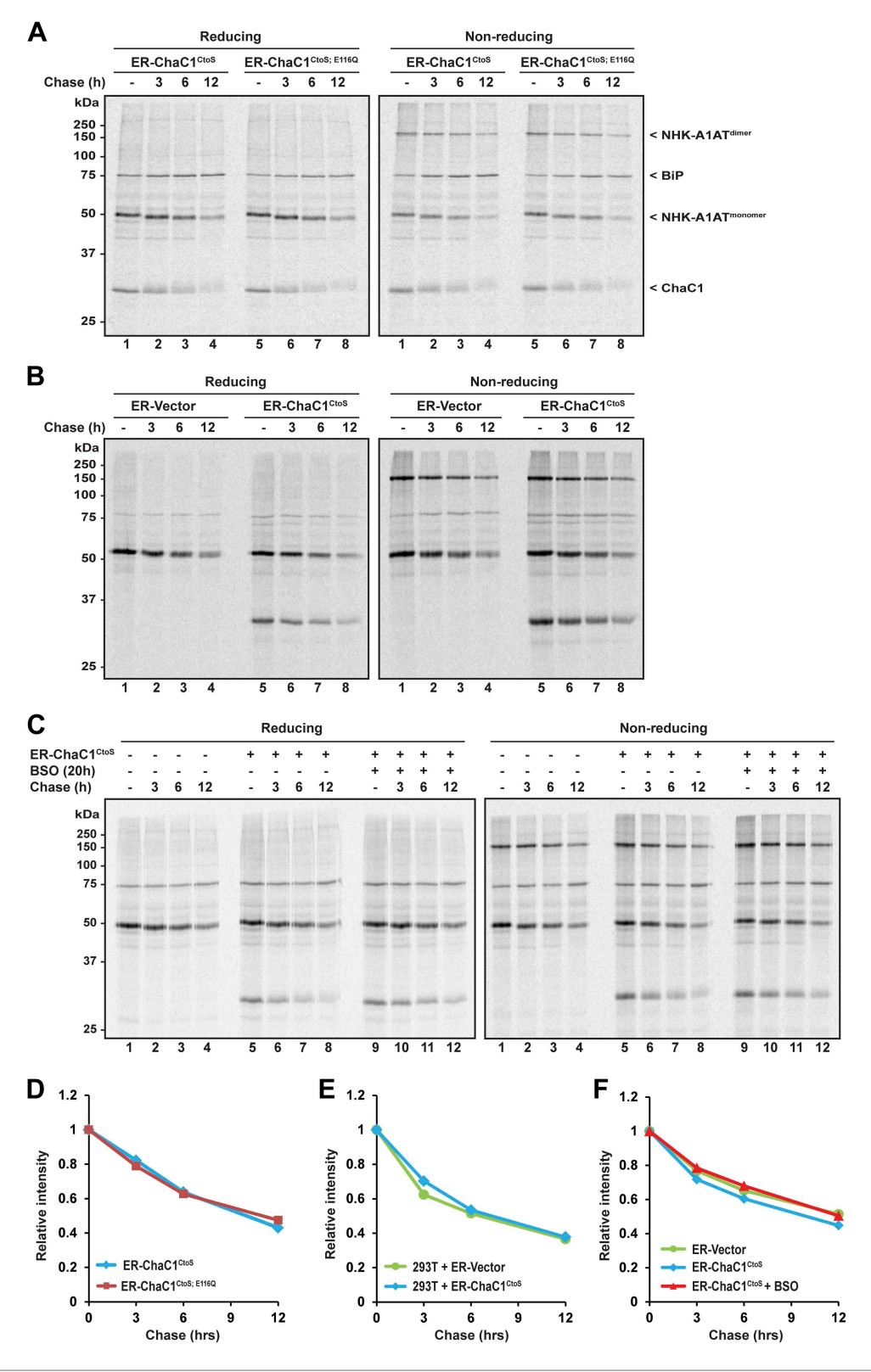

**Figure 5**. Clearance of misfolded null Hong Kong mutant alpha 1 anti-trypsin (NHK-A1AT) is unaffected by depletion of ER glutathione. (**A**) Autoradiograph of metabolically labeled NHK-A1AT immunopurified from HeLa cells co-expressing C-terminally FLAG-tagged NHK-A1AT and ER-localized enzymatically active (ER-ChaC1$^{CtoS}$) or

*Figure 5. Continued on next page*

*Figure 5. Continued*

inactive ChaC1 (ER-ChaC1$^{CtoS;E116Q}$) and resolved on a reducing or non-reducing SDS-PAGE. Cells were lysed at the end of a 30-min labeling pulse (lanes 1 and 5) or after an additional chase period (indicated). The mobility of the labeled NHK-A1AT, and the ER-chaperone BiP that co-purifies with it, are indicated, as is the labeled ChaC1, which is also recovered in this anti-FLAG immunoprecipitate. (**B**) Autoradiograph of samples recovered from 293T cells in an experimental design as in '**A**'. (**C**) Autoradiograph of samples recovered from HeLa cells in an experimental design as in '**A**'. Where indicated, cells were exposed to the glutathione synthesis inhibitor buthionine-sulfoxide (BSO, 100 µM, 20 hr) before the pulse-chase labeling. (**D–F**) Plot of time-dependent change in NHK-A1AT (monomer) signal from the reducing gels **A–C** above. Shown are representative experiments reproduced twice with similar outcome.

Further support for the idea of a shared economy of glutathione is provided by the phenotype of yeast over-expressing a plasma membrane glutathione transporter HGT1. Glutathione over-load in these cells compromises protein folding in the ER and triggers an unfolded protein response, the basis of which appears to be unregulated re-cycling of PDI from its oxidized to its reduced state imposed by the excessive glutathione (*Kumar et al., 2011*). While our state of ignorance in regard to the genes responsible for glutathione transport into the yeast ER preclude assigning a role to luminal glutathione in this consequences of cellular glutathione overload, the observations of Kumar et al. clearly argue that glutathione may participate in ER redox.

In mammalian cells too, glutathione stands to impact the maturation of some secreted or membrane proteins: In a study agnostic of its enzymatic activity, over-expression of wild-type cytosolic ChaC1 in mouse ganglionic eminence cells profoundly inhibited the maturation of the Notch precursor to its furin-cleaved form (*Chi et al., 2012*). Whereas anti-oxidants that restore impaired glutathione metabolism to normality have been shown to improve the capacity of liver cells to secrete factor VIII, a heavily disulfide bonded serum protein (*Malhotra et al., 2008*). Thus, conservatively interpreted, our observations lead to the conclusion that ER glutathione is not generally required to maintain protein thiol redox nor folding homeostasis in the ER of cultured mammalian cells. It may now be informative to examine the impact of an ER glutathione purge elicited by ER-localized ChaC1$^{CtoS}$ on the fate of specialized secretory cells and their specialized secretory cargo proteins.

## Materials and methods

### Plasmid construction

*Table 1* lists the plasmids used, their lab names, description, published reference, and a notation of their appearance in the figures.

The mouse ChaC1 cDNA (IMAGE clone 4483043) was purchased from Source Bioscience. Bacterial expression vectors encoding C-terminally His 6X tagged wild type and E116Q mutant ChaC1 were constructed by PCR amplification. A mutant ChaC1 in which all four cysteines were converted to serines, CtoS (C92S, C169S, C190S, C212S), was synthesized as an artificial gene and shuttled into the mammalian and bacterial expression plasmids described in *Table 1*.

Bacterial expression plasmids for roGFP2 (*Hanson et al., 2004*) and the glutaredoxin 1 fusion to roGFP2 (*Gutscher et al., 2008*) were gifts of James Remington (U Oregon) and Tobias Dick (DKFZ, Heidelebrg). ER localized mammalian expression plasmids of their counterparts were generated by deleting the E147b insertion and introducing an S65T mutation into ER_HA_Grx1_roGFP1iE_KDEL_pcDNA3.1 (*Birk et al., 2013*) (a gift of Christian Appenzeller-Herzog, University of Basel). ER_roGFP2 was produced by deleting E147b from FLAGM1_roGFP2_iE_pcDNA3.1 (*Avezov et al., 2013*).

Mammalian expression plasmids encoding ER localized FLAG M1-tagged ChaC1 with a KDEL ER retention signal at the C-terminus and mutant derivatives thereof were constructed by shuttling the ChaC1 coding sequence into the relevant plasmid backbone. Mammalian expression plasmids for ER-localized ChaC1 fused to the mCherry fluorescent protein were prepared using similar techniques.

The hyperactive C104A: C133A allele of human ERO1L (ERO1A) was expressed from a modified pCDNA3 plasmid in which the coding sequence of the Neo$^r$ marker had been replaced by human CD2.

### Protein purification and in vitro enzymatic assays

C-terminally 6X His-tagged mouse ChaC1 was expressed in BL21 pLysS *E. coli* at 30°C with 4 hr of induction with 1 mM IPTG and purified from the lysate by nickel affinity chromatography followed

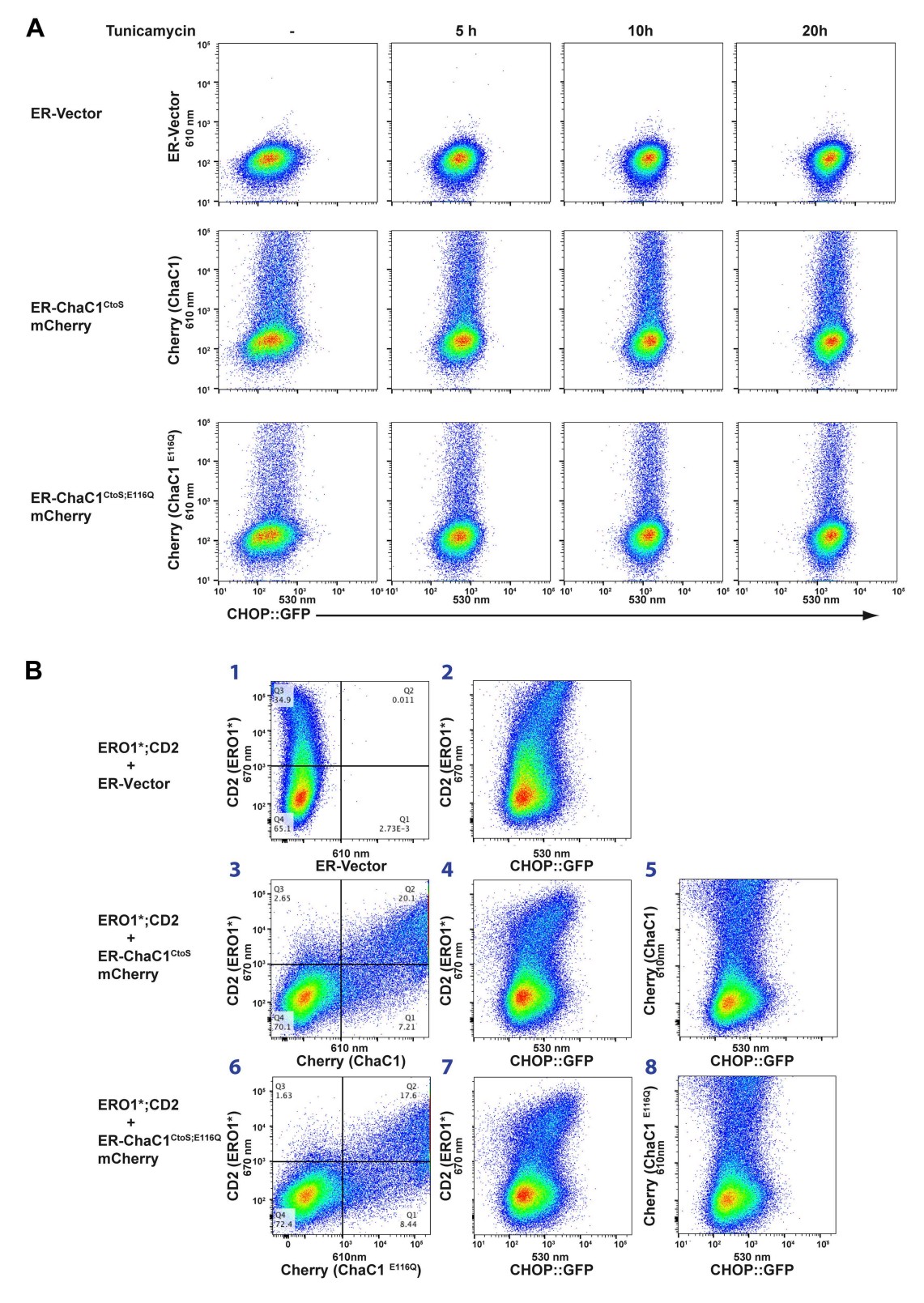

**Figure 6**. Activity of the unfolded protein response (UPR) is unaffected by depletion of ER glutathione. (**A**) Two-dimensional plots (FACScans) of fluorescent intensity of individual CHO cells containing a stably-integrated UPR reporter, CHOP::GFP (*Novoa et al., 2001*), following transfection with plasmids encoding ER-localized, active (ER-ChaC1^CtoS-mCherry-KDEL) or inactive ChaC1 (ER-ChaC1^CtoS;E116Q-mCherry-KDEL). GFP fluorescent intensity,
*Figure 6. Continued on next page*

*Figure 6. Continued*

reporting on the activity of the UPR (X-axis), was detected at 530 ± 30 nm following excitation at 488 nm, whereas mCherry fluorescent intensity, reporting on the level of ChaC1-mCherry-KDEL in the ER (Y axis), was detected at 610 ± 20 nm following excitation at 561 nm. Where indicated the cells were exposed to the ER stress-inducing agent tunicamycin for the indicated period of time. (**B**) Three-color FACScans as in '**A**' of CHOP::GFP cells co-transfected with expression plasmids for a hyperactive mutant of ERO1 (C104A, C133A; ERO1*) tagged by the cell-surface marker CD2 (decorated with an AF647-tagged antibody and detected by excitation at 640 nm and emission at 670 nm) with a wildtype or inactive mutants of ChaC1 fused to mCherry, as noted above. The axis of the scans are labeled with the cognate signals and panels numbered for ease of reference in the text.

The following figure supplements are available for figure 6:

**Figure supplement 1**. Activity of the unfolded protein response (UPR) as measured by an XBP1-splicing reporter is also unaffected by depletion of ER glutathione.

by size exclusion chromatography on Superdex 200 (GE Healthcare, Chalfont St Giles, United Kingdom) in 20 mM Tris pH7.4, 100 mM NaCl, 10% Glycerol, 1 mM Tris(2-carboxyethyl)phosphine hydrochloride (TCEP).

Glutathione degradation by purified wild-type ChaC1 and its mutant derivatives was assayed using the 5,5'-dithiobis-(2-nitrobenzoic acid) (DTNB) recycling assay of *Tietze (1969)* as modified by *Griffith (1980)*.

Mutant ChaC1 (C169S, C190S, C212S) fused at its C-terminus to an mCherry 6X His fluorescent protein was purified from *E. coli* and labeled at its single remaining cysteine (C92) with thiol reactive Oregon Green 488 iodoacetamide (Life Technologies), according to the manufacturer's instructions.

Fluorescent resonance energy transfer (FRET) between the Oregon Green donor and the mCherry acceptor was measured at room temperature on TECAN 500 plate reader as the ratio between their fluorescence emission at 535 nm and 670 nm respectively when they were excited at 485 nm and 590 nm. The binding reaction was initiated by addition of reduced glutathione (G4251, Sigma) or oxidized glutathione (G4626, Sigma).

Human PDI (PDIA1 18–508) and roGFP variants were expressed in the *E. coli* BL21 (DE3) strain, purified with Ni-NTA affinity chromatography, dialyzed into the reaction buffer, reduced by incubation with 20 mM of DTT, and then buffer exchanged on a PD-10 gel filtration column (GE Healthcare), as described previously (*Avezov et al., 2013*).

Reduced PDI (5 µM) was equilibrated in 100 mM Tris–HCl, 150 mM NaCl, pH 7.5 degassed buffer before being added to samples containing roGFP variants in the reduced state followed by oxidized glutathione (100 µM; Sigma). The ratio of fluorescence emission at 535 nm of samples sequentially excited at 405 nm and 488 nm was measured using a Synergy 4 microplate reader (BioTek Instruments). Kinetic parameters were extracted by fitting the data to a Linweaver–Burk plot.

## Cell culture, transfection

HeLa and HEK293T cells were cultured in DMEM supplemented with 10% FBS and maintained at 37°C with 5% $CO_2$. Cells were electroporated with the indicated plasmids (5 µg of DNA/1 × 10^6 cells) utilizing the Neon transfection system or Lipofectamine LTX (both from Life Technologies) following the manufacturer's protocols.

The effects of cytosolic and ER-localized ChaC1 on cellular glutathione pools were analyzed in stable clones of doxycycline-inducible Flp-In™ T-REx™ HEK 293T cells by the 5,5'-dithiobis-(2-nitrobenzoic acid) (DTNB) recycling assay of *Tietze (1969)* as modified by *Griffith (1980)*.

The effects of ChaC1 on the activity of the unfolded protein response were studied by transient transfection of stable CHOP::GFP reporter Chinese Hamster Ovary cells (C30 clone of CHO K1 cells [*Novoa et al., 2001*]) or a stable clone of CHO K1 cells (XV8-1) expressing a Venus reporter of XBP1 splicing activity (*Iwaki et al., 2004*) followed by multi-channel FACS analysis on an LSRFortessa (BD Bioscience) of the GFP or Venus signals, which reports on the intensity of the unfolded protein response and mCherry expression that reports on the level of expression of ER localized ChaC1. Where indicated, a CD2-marked expression plasmid encoding a hyper-active mutant of human ERO1L (C104A; C133A, *Hansen et al., 2012*) or CD2-marked empty vector was co-transfected and CD2 expression detected by surface staining with an AlexaFluor 647 conjugated mouse anti-human CD2 antibody, clone LT2 (MCA1194A647, AbD Serotec/Biorad).

**Table 1.** List of the plasmids used in this study, their unique lab identifier, lab name, description, PMID of the relevant reference (if available), figure in which they first appear and cognate label in figure legend

| ID | Plasmid name | Description | Reference | First appearance | Label in figure |
|----|----|----|----|----|----|
| 15 | pFLAG-CMV1 | Mammalian expression bovine trypsinogen signal peptide-FLAGM1 fusion | PMID: 8024796 | 2A | ER-FLAG_vector |
| 242 | roGFP2_pRSETB | Bacterial expression of 6X His-tagged roGFP2 | PMID: 14722062 | 3A | roGFP2 |
| 836 | mChac1_1-224-H6-pET30a | Bacterial expression of mouse Chac1 C-terminal His-tagged | This paper | 1A | ChaC1 |
| 888 | pFLAG_mCherry_KDEL_CMV1 | ER localised FLAGM1-mCherry-KDEL in pFLAG-CMV1 | This paper | 6A | ER-vector |
| 915 | mChaC1_1-224_CtoS_pET30a | Bacterial expression mouse Chac1 CtoS (C92S, C169S, C190S, C212S) C-terminal His-tagged | This paper | 2C | ChaC1_CtoS |
| 932 | mChaC1_1-224_E116Q_pET30a | Bacterial expression of E116Q mutant mouse Chac1 C-terminal His-tagged | This paper | 1E | ChaC1_E116Q |
| 934 | mChaC1_3XFLAG_pCDNA5_FRT_TO | Mammalian expression of C-term FLAG-tagged mouse ChaC1 | This paper | 3H | Cyto-ChaC1 |
| 937 | mChaC1_CtoS_E116_pET30a | Bacterial expression of E116Q cysteine to serine mutant mouse Chac1 C-terminal His-tagged | This paper | 2C | ChaC1_CtoS_E116Q |
| 945 | mChaC1_E116Q_3XFLAG_pCDNA5_FRT_TO | Mammalian expression of C-term FLAG-tagged mouse ChaC1 E116Q mutant (Cyto-ChaC1_E116Q) | This paper | 3H | Cyto-ChaC1_E116Q |
| 950 | FLAGM1_mChaC1_CtoS_pCDNA5_FRT_TO | Mammalian expression ER-localised FLAG M1 tagged mouse CHAC1 CtoS (C92S, C169S, C190S, C212S) KDEL | This paper | 2A | ER-FLAG_ChaC1_CtoS |
| 951 | FLAGM1_mChaC1_CtoS_E116Q_pCDNA5_FRT_TO | Mammalian expression ER-localised FLAG M1 tagged mouse CHAC1 CtoS (C92S, C169S, C190S, C212S) E116Q KDEL | This paper | 2A | ER-FLAG_ChaC1_CtoS_E116Q |
| 974 | FLAGM1_mChaC1_WT_pCDNA5_FRT_TO | Mammalian expression ER-localised FLAG M1 tagged mouse CHAC1 KDEL | This paper | 2A | ER-FLAG_ChaC1 |
| 988 | FLAGM1_mChaC1_CtoS_mCherry_pCDNA5_FRT_TO | Mammalian expression ER-localised FLAG M1 tagged mouse CHAC1 mCherry-KDEL, CtoS (C92S, C169S, C190S, C212S) | This paper | 3F | ER-ChaC1_CtoS_mCherry |
| 993 | mChaC1_CtoS_92C_mCherry-pET30a | Bacterial expression of mouse ChaC1-mCherry fusion, C-terminal His-tagged, (C169S, C190S, C212S) | This paper | 1D | OG-ChaC1-Cherry probe |
| 1028 | FLAGM1_mChaC1_CtoS_E116Q_mCherry_pCDNA5 | Mammalian expression ER-localised FLAG M1 tagged mouse CHAC1 mCherry-KDEL, CtoS, E116Q | This paper | 3F | ER-ChaC1_CtoS_E116Q_mCherry |

*Table 1. Continued on next page*

*Table 1. Continued*

| ID | Plasmid name | Description | Reference | First appearance | Label in figure |
|---|---|---|---|---|---|
| 1037 | mChaC1_CtoS_ S92C_E116Q_ mCherry-pET30a | Bacterial expression of mouse ChaC1-mCherry fusion, C-terminal His-tagged, (C169S, C190S, C212S), E116Q | This paper | S1C | OG-ChaC1_E116Q-Cherry |
| 1052 | ER_roGFP2_ pCDNA3.1 | ER localized roGFP2 KDEL | This paper | 3D | ER-roGFP2 |
| 1063 | ER_Grx1_roGFP2_ KDEL_pCDNA3.1 | ER localized Grx1 fused to roGFP2 KDEL | PMID:23424194 | 3D | ER-Grx1-roGFP2 |
| 1181 | hLDLR_3XFLAG_ pCDNA5_FRT | Mammalian expression plasmid of human LDL receptor, cytosolic tail tagged with a 3X FLAG tag | PMID:12493918 | 4A | FLAG-tagged LDL-R |
| 1204 | A1AT_NHK_ 3XFLAG_pCDNA5_ FRT_TO | Mammalian expression plasmid of null Hong-Kong mutant a1-antitrypsin C-terminally tagged with 3XFLAG | PMID:12736254 | 5A | FLAG-tagged NHK-A1AT |
| 1206 | Grx1_roGFP2_pET30a | Bacterial expression of Grx1-roGFP2 fusion protein | PMID: 18469822 | 3B | Grx1-roGFP2 |
| 1239 | pCAX-F-XBP1ΔDBD-venus | XBP1 mini-cDNA with Venus fused the post-IRE1 spliced open reading frame | PMID: 14702639 | 6C | XBP1-Venus |
| 1273 | hERO1A_C104A_ C131A_pCDNA3-CD2 | Mammalian expression plasmid encoding human hyperactive ERO1L (ERO1a) and a co-expressed human CD2 FACS marker | PMID: 23027870 | 6B | ERO1* |

## Immunoblot analysis

24 hr after transfection, cells were washed twice with PBS and lysed for 30 min on ice in 1% Triton X-100, 20 mM Tris–HCl (pH 7.4), 150 mM NaCl, 1 mM EDTA, 0.1 mM PMSF, 3–7 TIU/L aprotinin, and 20 mM N-ethylmaleimide (NEM). The cellular lysate was centrifuged at 15,000 ×$g$ for 15 min, and the supernatant was used for protein assay using a BCA protein assay reagent. Total proteins (30 µg) were separated on 12% SDS–polyacrylamide gels and electroblotted onto PVDF membrane. Primary antibodies to the FLAG tag (Sigma Cat #F1804, 1:1000 dilution) or a rabbit antiserum to bacterially expressed H6-tagged mouse ChaC1 (residues 1–224, lab number UC8166, 1:1000 dilution) followed by IR800 conjugated secondary antibody and by scanning on a Licor Odyssey scanner.

## Immunofluorescence

Transfected HeLa cells were grown on coverslips. 24 hr after transfection, the cells were washed with PBS, fixed with 4% paraformaldehyde for 30 min at room temperature, permeabilized in PBS containing 0.1% Triton X-100 for 30 min at room temperature, and blocked in 1% BSA in PBS for 30 min at room temperature. Anti-FLAG antibodies (1:1000 dilution in PBS) in combination with rabbit anti-calreticulin antibodies (a gift of Steven High, University of Manchester, 1:1000 dilution in PBS) followed by goat anti-mouse and goat anti-rabbit secondary antibodies conjugated with DyLight 543 and DyLight 488 (1:1000 dilution in PBS, Jackson ImmunoResearch Laboratories), respectively. Nuclei were counter stained with Hoechst 33,342 (2 µg/ml in PBS) for 30 min at room temperature for counter-staining.

## Confocal microscopy

Cells co-transfected with the redox reporter (roGFP) and ChaC1-mCherry were analyzed by laser-scanning confocal microscopy system (510 Meta; Carl Zeiss) with a Plan-Apo- chromat 63× oil immersion lens (NA 1.4), coupled to a microscope incubator, maintaining standard tissue culture conditions

(Okolab). Fluorescence ratiometric intensity images (512 × 512 points, 16 bit) of live cells were acquired. A diode 405 nm and Argon 488 nm lasers (2 and 0.5% output respectively) were used for excitation of the ratiometric probes in the multitrack mode with an HFT 488/405 beam splitter, the signal was detected with 518–550 nm filters, the detector gain was arbitrary adjusted to yield an intensity ratio of the two channels approximating one.

The recovery half-time was extracted from fitting the intensity ratio changes over time to an exponential equation $I(t) = A(1 - e^{-\tau t})$, where I is intensity, t is time, τ is recovery half-time.

## Metabolic labeling and pulse-chase analysis

Cells were co-transfected with ER_mChaC1 and the FLAG-tagged LDL receptor or mutant alpha1 antitrypsin NHK plasmids. 24 hr later, pulse-chase labeled, followed by anti-FLAG immunoprecipitation (Sigma A2220), SDS-PAGE and autoradiography were conducted as previously described (*Zito et al., 2010a*). Where indicated, the glutathione synthesis inhibitor buthionine-sulfoxide (BSO, 100 µM, 20 hr) was added before the pulse-chase labeling. Purified protein intensities were quantified using ImageJ software. Images of gels were scaled to fit the page dimensions.

## Acknowledgements

We are grateful to Tobias Dick and Christian Appenzeller-Herzog for the gift of ER_HA_Grx1_roGF-P1iE_KDEL_pcDNA3.1 and Grx1-roGFP2_pQE60. Jim Remington for the gift of the roGFP2 cDNA, Ineke Braakman for the gift of the LDL receptor cDNA, Nobuko Hosokawa for the gift of mutant a1-antitrypsin cDNA and Steve High, University of Manchester for the gift of the calreticulin antiserum.

Supported by a grant from the Wellcome Trust (Wellcome 084812/Z/08/Z) and the European Commission (EU FP7 Beta-Bat No: 277713) to DR, a Wellcome Trust Strategic Award for core facilities to the Cambridge Institute for Medical Research (Wellcome 100140) and the Fundação para a Ciência e Tecnologia, Portugal (project grant PTDC/QUI-BIQ/119677/2010) to EPM. DR is a Wellcome Trust Principal Research Fellow.

## Additional information

### Competing interests

DR: Reviewing editor, *eLife*. The other authors declare that no competing interests exist.

### Funding

| Funder | Grant reference number | Author |
| --- | --- | --- |
| Wellcome Trust | Wellcome 084812/Z/08/Z | David Ron |
| European Commission | EU FP7 Beta-Bat No: 277713 | David Ron |
| Wellcome Trust | Wellcome 100140 | David Ron |
| Fundação para a Ciência e a Tecnologia | PTDC/QUI-BIQ/119677/2010 | Eduardo Pinho Melo |

The funders had no role in study design, data collection and interpretation, or the decision to submit the work for publication.

### Author contributions

ST, Designed, executed and interpreted the cell biological experiments shown in *Figures 2, 4, 5 and 6* and the in vivo redox measurements of *Figures 3 and 4*; EA, Oversaw the design, execution and interpretation of the fluorescent based measurements in *Figures 1, 3 and 4*; AZ, Designed, executed, interpreted and led the biochemical and biophysical experiments characterizing ChaC1 in vitro, shown in *Figure 1* and *Figure 1—figure supplement 1*; TK, Contributed to the design, execution, and interpretation of the in vivo redox measurements in *Figures 3 & 4*; LM-S, EPM, Contributed to the design, execution, and interpretation of the in vitro redox measurements of *Figure 3*; HPH, Oversaw and contributed to the design and interpretation of in vitro measurements of ChaC1's biochemical properties (*Figure 1*) and to execution of some of the cell biological experiments (*Figures 2 & 6*) as well as to the writing and editing of the manuscript; DR, Conceived and oversaw the study as a whole, wrote the manuscript, designed and constructed expression plasmids for the study

## Additional files

### Major dataset

The following previously published dataset was used:

| Author(s) | Year | Dataset title | Dataset ID and/or URL | Database, license, and accessibility information |
|-----------|------|---------------|----------------------|-------------------------------------------------|
| Oakley AJ, Yamada T, Liu D, Coggan M, Clark AG, Board PG | 2008 | Gamma-glutamyl cyclotransferase | http://www.pdb.org/pdb/explore/explore.do?structureId=2rbh | Publicly available at RCSB Protein Data Bank. |

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
