## [Decision Letter]

Thank you for sending your work entitled “Intact protein folding in the glutathione-depleted endoplasmic reticulum implicates alternative protein thiol reductants” for consideration at *eLife*. Your article has been favorably evaluated by Vivek Malhotra (Senior editor) and 3 reviewers, one of whom is a member of our Board of Reviewing Editors.

The Reviewing editor and the other reviewers discussed their comments before we reached this decision, and the Reviewing editor has assembled the following comments to help you prepare a revised submission.

It is widely accepted that reduced glutathione in the ER is the primary driver of the disulfide reduction needed for thiol editing as proteins mature. This paper uses an elegant approach to test this assumption directly. By expressing specifically in the ER a glutathione-degrading enzyme the authors assess the effect on various well established assays of folding, maturation and quality control. Surprisingly, none of the processes examined were affected by the loss of glutathione.

This represents a radical challenge to the current models. The experiments are thorough and carefully controlled. The authors discuss the implications carefully and quite conservatively but are forced to conclude that the presence of a glutathione-driven enzyme system in the ER lumen is not necessary for at least much of the protein folding and redox homeostasis in this organelle.

All three reviewers found this paper to be provocative, well done, and rigorous. The data are of outstanding quality, and they are succinctly and clearly described and discussed. There was a clear consensus that it was appropriate for *eLife* with some relatively minor amendments, outlined below.

The experiments and controls are logical but some of them are quite complex. Diagrams of the experimental logic might help in some cases.

Nomenclature should be standardized. Glutathione vs. GSSG vs. tripeptide (the latter being a bit jargony, unless explained). When Figure 6 is described in the text the ER-ChaC1-CtoS has lost the 'CtoS'.

Related to the data in Figure 6, depletion of total cellular glutathione with BSO has been described previously as aggravating the toxicity of ERO1* (13). The authors should offer some discussion as to how this prior observation can be reconciled with their conclusion that ER-ChaC1 has no impact on ER homeostasis in cells expressing ERO1* as measured by UPR induction.

It would be helpful to further explain the FACScans in Figure 6. For example, a few basic sentences of explanation such as “a shift to the right for the GFP signal depicts UPR induction” would be helpful. In addition, what do the boxes denote in Figure 6?

---

## [Author Response]

*The experiments and controls are logical but some of them are quite complex. Diagrams of the experimental logic might help in some cases*.

To help guide the reader through some of the less obvious experiments, we have revised the text to provide clear predictions on the correspondence between hypothesis and results. Furthermore, we have introduced a new panel into Figure 3 (panel A) cartooning the different kinetic coupling of the two redox probes (roGFP2 and Grx1-roGFP2) with the glutathione buffer.

*Nomenclature should be standardized. Glutathione vs. GSSG vs. tripeptide (the latter being a bit jargony, unless explained). When*
Figure 6
*is described in the text the ER-ChaC1-CtoS has lost the 'CtoS'*.

Glutathione is now uniformly referred to as such.

*Related to the data in*
Figure 6*, depletion of total cellular glutathione with BSO has been described previously as aggravating the toxicity of ERO1* (*[13]*). The authors should offer some discussion as to how this prior observation can be reconciled with their conclusion that ER-ChaC1 has no impact on ER homeostasis in cells expressing ERO1* as measured by UPR induction*.

We thank the reviewers for alerting us to the need to address this issue. In the revised version the Discussion cites Hansen’s findings and places them in context with ours.

*It would be helpful to further explain the FACScans in*
Figure 6*. For example, a few basic sentences of explanation such as “a shift to the right for the GFP signal depicts UPR induction” would be helpful. In addition, what do the boxes denote in*
Figure 6?

The explanation of the FACScan has been expanded. The unhelpful boxes were removed.